# Validation of N Protein Antibodies to Diagnose Previous SARS-CoV-2 Infection in a Large Cohort of Healthcare Workers: Use of Roche Elecsys^®^ Immunoassay in the S Protein Vaccination Era

**DOI:** 10.3390/v15040930

**Published:** 2023-04-07

**Authors:** Juan Francisco Delgado, Mònica Vidal, Germà Julià, Gema Navarro, Rosa María Serrano, Eva van den Eynde, Marta Navarro, Joan Calvet, Jordi Gratacós, Mateu Espasa, Pilar Peña

**Affiliations:** 1Immunology Laboratory, Clinic Laboratories Service, Parc Taulí Hospital Universitari, Institut d’Investigació i Innovació Parc Taulí (I3PT-CERCA), Departament de Medicina, Universitat Autònoma de Barcelona, 8207 Sabadell, Spain; 2Epidemiology Service, Parc Taulí Hospital Universitari, Institut d’Investigació i Innovació Parc Taulí (I3PT-CERCA), Universitat Autònoma de Barcelona, 8207 Sabadell, Spain; 3Occupational Health Department, Parc Taulí Hospital Universitari, Institut d’Investigació i Innovació Parc Taulí (I3PT-CERCA), Universitat Autònoma de Barcelona, 8207 Sabadell, Spain; 4Infection Disease Department, Parc Taulí Hospital Universitari, Institut d’Investigació i Innovació Parc Taulí (I3PT-CERCA), Universitat Autònoma de Barcelona, 8207 Sabadell, Spain; 5Rheumatology Service, Parc Taulí Hospital Universitari, Institut d’Investigació i Innovació Parc Taulí (I3PT-CERCA), Departament de Medicina, Universitat Autònoma de Barcelona, 8207 Sabadell, Spain; 6Microbiology Section, Laboratory Service, Parc Taulí Hospital Universitari, Institut d’Investigació i Innovació Parc Taulí (I3PT-CERCA), Universitat Autònoma de Barcelona, 8207 Sabadell, Spain

**Keywords:** SARS-CoV-2, antibody response, infection, vaccination, nucleocapsid protein, spike protein

## Abstract

The aim of this study was to validate the detection of anti-nucleocapsid protein (N protein) antibodies for the diagnosis of SARS-CoV-2 infection in light of the fact that most COVID-19 vaccines use the spike (S) protein as the antigen. Here, 3550 healthcare workers (HCWs) were enrolled from May 2020 (when no S protein vaccines were available). We defined SARS-CoV-2 infection if HCWs were found to be positive by RT-PCR or found to be positive in at least two different serological immunoassays. Serum samples from Biobanc I3PT-CERCA were analyzed by Roche Elecsys^®^ (N protein) and Vircell IgG (N and S proteins) immunoassays. Discordant samples were reanalyzed with other commercial immunoassays. Roche Elecsys^®^ showed the positivity of 539 (15.2%) HCWs, 664 (18.7%) were found to be positive by Vircell IgG immunoassays, and 164 samples (4.6%) showed discrepant results. According to our SARS-CoV-2 infection criteria, 563 HCWs had SARS-CoV-2 infection. The Roche Elecsys^®^ immunoassay has a sensitivity, specificity, accuracy, and concordance with the presence of infection of 94.7%, 99.8%, 99.3%, and 0.96, respectively. Similar results were observed in a validation cohort of vaccinated HCWs. We conclude that the Roche Elecsys^®^ SARS-CoV-2 N protein immunoassay demonstrated good performance in diagnosing previous SARS-CoV-2 infection in a large cohort of HCWs.

## 1. Introduction

COVID-19 is an acute respiratory syndrome caused by the new coronavirus SARS-CoV-2, first described in Wuhan (Hubei province, China) following an outbreak of pneumonia of unknown origin. It is highly transmissible and has spread throughout the world. The WHO declared its spread a pandemic causing COVID-19 [1].

Clinical manifestations of SARS-CoV-2 infection range from asymptomatic or mild non-specific symptoms to severe pneumonia with organ function damage. Common symptoms are fever, cough, fatigue, dyspnea, myalgia, sputum production, and headache [2,3]. These symptoms are non-specific and cannot be used for an accurate diagnosis; therefore, laboratory testing plays an important role in diagnosing SARS-CoV-2 patients. These tests can also identify those who are asymptomatic.

Laboratory diagnosis of COVID-19 has mainly been based on molecular tests such as real-time reverse-transcription PCR (RT-PCR) [4,5,6]. Antibody-based techniques are complementary tools for SARS-CoV-2 infection detection. The presence of antibodies is an indirect marker of infection [4,6,7,8,9,10]. The development of an antibody response to COVID-19 occurs between 5 and 14 days after exposure to the virus. As such, serological tests in the market are of little use in the context of acute COVID-19. Sensitivities are less than 50% in the first week of infection [11]. However, the detection of SARS-CoV-2 antibodies is an excellent way to determined past infection with a sensitivity higher than 90% after 7 days [9,12,13,14]. Serological assays play an essential role in population seroprevalence evaluation and can help to account for asymptomatic cases, symptomatic cases that did not get tested, or patients suspected to have COVID-19 with a negative SARS-CoV-2 RT-PCR.

SARS-CoV-2 has at least four structural proteins: spike (S), envelope (E), membrane (M), and nucleocapsid (N) proteins. Both viral S and N proteins are major structural proteins and highly immunogenic. Therefore, most patients develop antibodies against them. [15]. In addition, the SARS-CoV-2 genome encodes 16 nonstructural proteins [16]. Antibodies against peptides derived from non-structural and accessory proteins are also detectable [17]. Commercial serologic methods target specific antibodies on several SARS-CoV-2 epitopes including the N protein, the S protein, and the receptor-binding domain of the S protein. The tests provide accurate diagnosis if performed on specimens collected 10 to 14 days after symptom onset, but performance varies among methods [9,13]. Different studies relate antibody titers after SARS-CoV-2 infection with age, sex, and severity [18,19,20,21]. More than 350 vaccines are currently being investigated for a potential role in mitigating the COVID-19 pandemic (https://www.who.int/publications/m/item/draft-landscape-of-COVID-19-candidate-vaccines, accessed on 27 February 2023). The approved vaccines target the S protein because this is the one that binds to the ACE2 (angiotensin-converting enzyme 2) receptor; thus, developing a humoral immune response against it could generate the formation of neutralizing antibodies to prevent infection.

Several studies have analyzed the antibody response induced by the S protein [22]. In a healthy population, people develop an antibody response from mRNA vaccines (BNT162b2 and mRNA-1273). These antibodies act against the S protein of the original strain [23,24]. mRNA vaccines and other vaccines can induce this response against the S protein [25]. Global vaccination rates range from 40–90% depending on the country [26]. This makes the serological diagnosis of SARS-CoV-2 infection difficult because it is impossible to distinguish antibodies against the S protein by infection vs. those produced by vaccination. It is relevant to study antibodies against other virus proteins for serological diagnosis; the most widely used immunoassay assesses the N protein. [23,24,27].

We present here a study analyzing the performance of anti-SARS-CoV-2 IgM/IgA/IgG Elecsys^®^ (Roche Diagnostics International Ltd., Rotkreuz, Switzerland) on Cobas^TM^ e801 (Roche Diagnostics International Ltd., Rotkreuz, Switzerland) to detect N protein antibodies for diagnosing SARS-CoV-2 infection in 3550 healthcare workers (HCWs) during the first COVID-19 wave. We then compared its performance with the Vircell IgG immunoassay (Vircell, Granada, Spain), which detects peptides from N and S proteins and the RT-PCR.

## 2. Materials and Methods

This observational retrospective study was approved by the Drug Research Ethics Committee of Parc Taulí University Hospital (code 2020581).

### 2.1. Study Population

Here, 3550 HCWs were enrolled from 6 to 29 May 2020 when S protein vaccines were not available. The demographic and clinical characteristics of this study cohort are shown in Table 1, and tests performed in this cohort are shown in Figure 1. Inclusion criteria were HCWs from Parc Taulí University Hospital and HCWs who worked at the center during the pandemic. Clinical characteristics were obtained from a survey at the time of enrollment. The HCWs had to record their symptoms from the start of the pandemic in our area (at the end of February 2020) until the time of study participation. The HCWs’ serum samples were provided by BioBanc I3PT-CERCA and were processed after standard operating procedures with the approval of the Ethics and Scientific Committees for serological commercial immunoassays.

A validation cohort was designed using a new cross-sectional study of serologies in HCWs vaccinated with mRNA vaccines (BNT162b2 and mRNA-1273) and HCWs who had had a RT-PCR test. HCWs were classified as infected in the study cohort were excluded. Thus, the validation cohort was made up of 297 infected and 1593 non-infected HCWs according to RT-PCR results. The female/male ratio, age, body mass index and smoking variables were recorded using an online survey at the time of enrollment. These clinical data are shown in Appendix A.

### 2.2. SARS-CoV-2 Infection Criteria

We confirmed SARS-CoV-2 infection if HCWs were found to be positive by the RT-PCR or were found to be positive in at least two different serological immunoassays. The equivocal results of the immunoassays were interpreted as negative results when assessing the criteria. Serology is a widely known tool for the diagnosis of previous infectious diseases including COVID-19 [13]; however, the use of these infection criteria may have induced a bias when studying the performance of serological immunoassays in the study population, since a large number of HCWs were classified as infected and non-infected using serological tests exclusively, and this variable is part of the infection criteria. We limited the effect of this bias in two ways: by using the existence of two serological tests and not just one to classify a HCW as infected and performing 2 additional serological tests in case of discrepant results.

### 2.3. RT-PCR

The microbiological diagnosis of SARS-CoV-2 infection was carried out by nucleic acid amplification techniques. All patients had a nasopharyngeal swab and were tested for SARS-CoV-2 infection through a retrotranscriptase PCR (RT-PCR): swabs were processed by the RT-PCR for SARS-CoV-2 with Allplex 2019-nCoV Assay (RP10244Y, Seegene, Seoul, Republic of Korea) or the Simplexa SARS-CoV2 Assay kit (MOL4150, DiaSorin, Gerenzano, Italy) per the manufacturer’s instructions for qualitative results.

### 2.4. Commercial Immunoassays to Detect Antibodies against SARS-CoV-2

All serum samples were thawed to perform the Elecsys^®^ Anti-SARS-CoV-2 IgM/IgA/IgG assay on Cobas^TM^ e801 (09203079190, Roche Diagnostics International Ltd., Rotkreuz, Switzerland) according to manufacturer instructions. Samples were positive if the index was ≥1. The ELISA COVID-19 IgG immunoassay (G1032, Vircell, Granada, Spain) used Triturus^®^ ELISA Instrument (Grifols, Barcelona, Spain) according to the Vircell-adapted protocol for this analyzer; samples were positive if the index was ≥11.2. Antigens and immunoassay characteristics are shown in Appendix A.

Discordant samples between the two immunoassays were reanalyzed with other commercial immunoassays according to manufacturer instructions or by adapting its protocols to Triturus^®^ ELISA Instrument: ELISA Anti-SARS-CoV-2 (IgG) (EI 2606-9601 G, Euroimmun, Lubeck, Germany); LIAISON^®^ SARS-CoV-2 S1/S2 IgG (311450, DiaSorin, Gerenzano, Italy). Antigens and immunoassay characteristics are shown in Appendix A.

### 2.5. Validation Specificity of Immunoassays

In addition, 100 serum samples from healthy donors were collected from Banc de Sang i Teixits in a pre-pandemic period (October 2019) to establish a specificity of > 98% for the Roche Elecsys^®^ and Vircell IgG immunoassays. The healthy donors were aged between 18 and 69 years, including 54 males and 46 women.

### 2.6. Statistical Analysis

For descriptive purposes, the cohort was characterized with absolute and relative frequencies for categorical variables; medians were used for numerical measurements. Sensitivity, specificity, positive, negative predictive values, and area under the receiver operating characteristic (ROC) curve were calculated for Roche Elecsys^®^ and Vircell IgG immunoassays. Level of agreement with SARS-CoV-2 infection was calculated with Cohen’s kappa coefficient. The Kolmogorov–Smirnov test was used to evaluate the suitability of data for normal distribution. We used univariate analysis to test the link between variables with the Chi square test or Fisher’s exact test if indicated for categorical variables. A Mann–Whitney U-test was used for continuous quantitative variables. Significant associations were assumed when *p* was <0.05. Analysis used the statistical software IBM SPSS Statistics v28.0, (Chicago, IL, USA).

## 3. Results

### 3.1. Immunoassays’ Specificity

Specificity was analyzed with serum samples from 100 healthy donors. None of the processed samples gave a positive result on the Roche Elecsys^®^ immunoassay within the index cutoff established by the manufacturer (1.0). However, five samples were positive for IgG for the Vircell IgG immunoassay within the manufacturer’s recommended index cutoff (6.0). The index cutoff for Vircell IgG was 11.2 for a specificity of 98%.

### 3.2. Immunoassays’ Performance

Samples from 3550 HCWs were tested by Roche Elecsys^®^ and Vircell IgG immunoassays: 539 (15.2%) and 664 (18.7%) were found to be positive by each immunoassay, respectively. There were 164 samples (4.6%) with discrepant results between immunoassays (Figure 2). To determine the presence of SARS-CoV-2 antibodies, discrepant samples were tested with Euroimmun IgG and DiaSorin IgG immunoassays. Concordance between immunoassays for 164 discrepant samples are shown in Table 2. Among the 164 discrepant samples, 23 were found to be positive by Roche Elecsys^®^, and 14 were also found to be positive by both Euroimmun IgG and Diasorin IgG. There were 141 samples found to be positive by Vircell IgG; four of them were found to be positive by Euroimmun IgG, and seven of them were found to be positive by Diasorin IgG. Therefore, in this group, Roche Elecsys^®^ showed a positive concordance with immunoassays, except for Vircell IgG, and with SARS-CoV-2 infection. Vircell IgG showed discordance with the Roche Elecsys^®^ and DiaSorin IgG immunoassays and infection. Therefore, the Roche Elecsys^®^ immunoassay showed better performance on discrepant samples.

### 3.3. RT-PCR Performance

A total of 425 HCWs were tested for SARS-CoV-2 by RT-PCR; 203 (47.8%) gave positive results between the start of the pandemic (February 2020) and the time of enrollment in the study (May 2020). In the group of HCWs with positive RT-PCR results, the Roche Elecsys^®^ and Vircel IgG immunoassay detected antibodies in almost 90% (Table 3). However, concordance between the RT-PCR and serology dropped significantly among HCWs with negative RT-PCR results—antibodies against SARS-CoV-2 were detected in 30% of the HCWs (Table 3). The detection of antibodies using the Roche Elecsys^®^ immunoassay showed a better correlation with SARS-CoV-2 infection than Vircell IgG did among the group of HCWs with RT-PCR results (Table 3).

### 3.4. N protein Antibody Immunoassay Performed to Diagnose SARS-CoV-2 Infection

According to our SARS-CoV-2 infection criteria, 563 among the 3550 HCWs in this study had SARS-CoV-2 infection. Among the infected HCWs, the immunoassay detecting N protein antibodies, Roche Elecsys^®^ immunoassay, was positive in 534 samples (94.8%) and Vircell IgG was positive in 523 (92.9%) samples (Table 4). The sensitivity, specificity, negative predicted value, positive predicted value and accuracy of both immunoassays are shown in Table 4. The N protein antibodies detected by Roche Elecsys^®^ showed excellent correlation and accuracy with SARS-CoV-2 infection—these metrics were higher than those found using the Vircell IgG immunoassay (Table 4).

### 3.5. Antibodies against SARS-CoV-2 and RT-PCR Association with Clinical Symptoms

As expected, HCWs with SARS-CoV-2 and hospitalized infected groups had symptoms more frequently associated with infection except for arterial hypertension (Table 1). The infected HCW group had younger workers and fewer smokers than the non-infected HCW group dod (Table 1). Given the strong correlation between antibody detection and infection, the association between antibodies and the presence of symptoms has similar statistical significance (Appendix A). Of the HCWs with RT-PCR results, the infected HCWs with positive and negative RT-PCR results were compared. Both groups showed similar behavior except HCWs who were overweight or had a dry cough or comorbidities. Differences were observed in the percentage of asymptomatic patients between both groups and the frequency of positive antibodies found by Roche Elecsys^®^ and Vircell IgG. There were more positive antibodies in HCWs with negative RT-PCR results (Appendix A). Finally, the performance of the immunoassays for the detection of antibodies against SARS-CoV-2 in symptomatic HCWs were compared with that in asymptomatic HCWs: no significant differences were found between Roche Elecsys^®^ and Vircell IgG at a quantitative level (*p* = 0.243; *p* = 0.629, respectively) or a qualitative level (*p* = 0.206; *p* = 0.687, respectively). One difference between symptomatic and asymptomatic HCWs was seen between smokers (18.1%) in the groups of asymptomatic vs. symptomatic subjects (8.7%), *p* = 0.001.

### 3.6. Roche Elecsys^®^ Immunoassay Performance in the Validation Cohort

Finally, we analyzed the performance of the Roche Elecsys^®^ immunoassay in the validation cohort—a cohort of vaccinated HCWs with a positive result found by the RT-PCR who did not belong to the study cohort. Sensitivity, specificity, negative predicted value, positive predicted value, and accuracy are shown in Table 5. The results obtained by the immunoassay in the validation cohort were very similar to those obtained in the study cohort. In this cohort, infected HCWs were younger than non-infected HCWs, at ages of 43.0 (IQR 34.0–52.0) versus 45.0 (37.0–54.0), respectively (*p* = 0.007). Smoking was more frequent in the non-infected HCWs *p* < 0.001 (Appendix A). There were no significant differences in sex and body mass index between the infected and non-infected HCWs (Appendix A).

## 4. Discussion

This study analyzed the performance of the Roche Elecsys^®^ immunoassay directed against the SARS-CoV-2 N protein in a large cohort of HCWs (*n* = 3550). A positive RT-PCR and/or two positive serological tests were the infection criteria. All samples were collected in May 2020 before the approval of S protein vaccines.

The Roche Elecsys^®^ test measures antibodies against the N protein. The results were positive for 539 HCWs. The Vircell IgG test measures antibodies against S protein and N protein peptides and was positive for 664 HCWs. The overall seroprevalence in our cohort was 15.2% (539) for Roche Elecsys^®^ and 18.7% (664) for Vircell IgG in May 2020. Initially, the higher antibody positivity found by Vircell IgG assay could have been due to how it analyzes S and N proteins. The Roche Elecsys^®^ only analyzes N proteins as reported in a meta-analysis [28]. As such, we decided to evaluate the 164 discrepant samples with DiaSorin IgG and Euroimmun IgG immunoassays. Concordance between the Vircell IgG test and the new tests confirmed that the differences were discordant, ranging between −0.18–0.12 (Cohen’s kappa coefficient). In Roche’s test, however, the concordance was moderate, ranging between 0.58–0.61 (Cohen’s kappa coefficient) (Table 2). A possible explanation is the immunoglobulin isotype that each immunoassay detects. Roche Elecsys^®^ measures IgG, IgM, and IgA while Vircel IgG only measures IgG. However, DiaSorin IgG and Euroimmun IgG only detect the IgG isotype and show better performance than Vircell IgG. Thus, the apparent increased sensitivity of the Vircell IgG immunoassay is translated into a loss of specificity and lower positive predictive value (Table 4). In this group of discrepant samples, DiaSorin IgG, Roche Elecsys^®^, and Euroimmun IgG showed good correlation with SARS-CoV-2 infection (Table 2). However, the Euroimmun and DiaSorin immunoassays use the S protein as an antigen; therefore, they would not be useful for the diagnosis of infection, while the Roche Elecsys^®^ would have this utility given the worldwide COVID-19 vaccination.

May 2020 was a period of the first wave of the pandemic and there was limited access to RT-PCR; thus, not all HCWs with symptoms underwent RT-PCRs during acute infection with SARS-CoV-2. The lack of availability of reagents for the diagnosis of acute infection by RT-PCR causes it to have a low yield with a Cohen’s kappa coefficient of 0.68 (Table 3). The main cause was the time between the test’s performance and the onset of symptoms. In many cases, this was over two weeks, thus giving negative RT-PCR results; however, 67 of the 202 HCWs found to be negative by RT-PCR were classified as infected due to the positivity of two serological tests. Thus, we evaluated the relevance of serological tests for the diagnosis of past infection.

Among the group of RT-PCR-positive HCWs, around 90% had seroconverted; the infection criterion we used assumes that a positive RT-PCR indicates infection without accounting for the false positive rate found by this technique. The sensitivity of both serologic immunoassays could be slightly underestimated (Table 4) [29]. Among the HCWs, Roche Elecsys^®^ obtained the best Cohen’s kappa coefficient (Table 3). Of the RT-PCR-negative HCWs, around 30% had positive antibodies: This could be explained by the lack of reagents—some PCRs were performed outside the period of acute infection, when its sensitivity decreased [30].

The Roche Elecsys^®^ immunoassay was better in terms of sensitivity, specificity, accuracy, and concordance with infection with values of 94.7, 99.8, 99.3, and 0.96, respectively, in the study cohort. The Vircell IgG assay was found to have the same values, 93.0, 95.3, 96.9, and 0.83, respectively. If we used the cutoff point recommended by Vircell for analysis, then specificity would have dropped further and the assay would have had worse results. This is in contrast to the data from Alharbi et al. [31]. The improved performance of the Roche Elecsys^®^ immunoassay agrees with the results of different comparative studies using a small sample size. This research confirms the results obtained by other studies using a large cohort of patients [12,32,33,34]. Additionally, the Roche Elecsys^®^ immunoassay was analyzed in a validation cohort consisting of vaccinated HCWs who were classified as infected HCWs using RT-PCR test results. In this cohort, the immunoassay showed a sensitivity, specificity, accuracy, and concordance with infection of 95.3, 99.7, 96.9, and 0.96, respectively. The performance of Roche Elecsys^®^ in both study and validation cohorts showed similar results; therefore, the possible bias that the analysis of the performance of the serological test could have as it is also part of the classification criteria was minimized by the correction factors included to reduce this effect.

As expected, symptoms related to SARS-CoV-2 infection were more frequently present in infected HCWs than non-infected HCWs. There are three risk factors associated with severe infection that HCWs in our group did not have: arterial hypertension, obesity, and age (Table 1). This is probably because the population chosen for the study was young with little associated comorbidities compared to the general population. The presence of arterial hypertension was not associated with infection since it is a risk factor for those suffering from more serious diseases than SARS-CoV-2. The mechanism is that treatments with angiotensin-converting enzyme inhibitors induce the upregulation of ACE2, which is the receptor that the virus uses to enter inside cells [35,36,37]. Obesity was another risk factor evaluated in the survey that was not associated with infection [38,39]. The last risk factor not associated was age—this association was the opposite of what was expected. Infection was more frequent in younger HCWs [40,41]. Another fact is that the smoking rate in non-infected HCWs was higher than that in infected ones. It was reported that infected patients who smoke have a lower antibody response, which is a risk factor for severe SARS-CoV-2 infection; the frequency in hospitalized patients was lower than in the normal population [24,42,43]. In our cohort of HCWs, we saw a paradox of smokers being in the non-infected HCWs group. This fact could be explained due to the bias of the population studied. The clinical variables recorded in the validation cohort showed similar associations between them compared to the study cohort, where infected HCWs were younger and showed a lower smoking rate than non-infected HCWs.

We compared infected HCWs under our infection criteria based on their RT-PCR positivity. We determined if HCWs could be classified as infected exclusively by performing two positive serological tests, which presented a difference with respect to those who had been diagnosed with a positive RT-PCR. In comparison, we observed that most clinical parameters were similar in both groups. Differences were found in terms of the higher frequency of overweight, comorbid, and asymptomatic HCWs between the group diagnosed by serology and the HCW group diagnosed with positive RT-PCRs. There was a higher frequency of dry cough (Table 5). The presence of asymptomatic infection for HCWs diagnosed by serology is explained by the fact that RT-PCR were more likely to be performed during the period of acute infection when symptoms were present than on asymptomatic HCWs, RT-PCR results of whom indicated if there was close contact with a positive case.

Lastly, we analyzed whether asymptomatic HCWs had a different antibody response to infection than HCWs with symptoms; there were no significant differences found between Roche Elecsys^®^ and Vircell IgG (*p* = 0.243 and *p* = 0.629, respectively). There were no differences in sex, age, and weight. We found a higher frequency of HCWs who were smokers in the asymptomatic group vs. the symptomatic group (*p* = 0.001).

## 5. Conclusions

The Roche Elecsys^®^ SARS-CoV-2 N protein immunoassay was used in a large cohort of HCWs and demonstrated good performance in the diagnosis of previous infection SARS-CoV-2. It is a useful tool to diagnose previous infection in a population vaccinated against SARS-CoV-2 via the S protein.

## Figures and Tables

**Figure 1 viruses-15-00930-f001:**
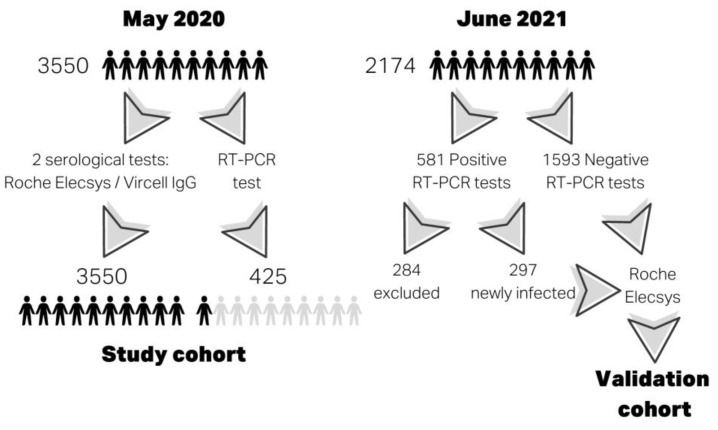
Definition of cohorts and tests performed.

**Figure 2 viruses-15-00930-f002:**
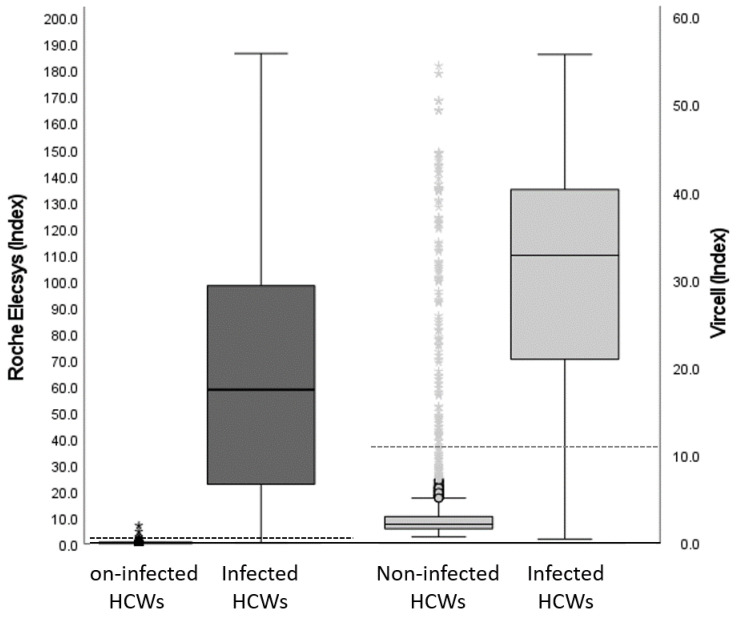
Antibody response against SARS-CoV-2 nucleocapsid protein in the study cohort of health care workers; Roche Elecsys^®^ on the left and Vircell IgG on the right. Subjects were grouped according to their infection status. Boxplots represent the quantification of the distribution of anti-nucleocapsid protein antibodies; upper and lower bounds of the boxes indicate 75th and 25th percentiles, respectively. The dotted horizontal line indicates the cut-off point of each immunoassay.

**Table 1 viruses-15-00930-t001:** Clinical data of healthcare workers from the study cohort (*n* = 3550).

		HCW SARS-CoV-2	HCW without SARS-CoV-2	*p* Value
		Infection (*n* = 563)	Infection (*n* = 2987)	
Clinical	Age in years (median ± IQR)	39.0 (29.0–50.0)	42.0 (33.0–52.0)	<0.001
	Female/Male ratio	3.8	3.5	0.534
characteristics	Days after onset symptoms (median ± IQR)	55.0 (44.0–54.0)	57.0 (41.0–70.7)	0.041
	Body mass index (median ± IQR)	23.5 (21.5–26.6)	23.9 (21.5–26.7)	0.573
	Overweight (%)	27.4	28.7	0.520
	Obese (%)	9.4	10.0	0.665
	Smoker (%)	11.5	25.1	<0.001
	Hospitalization (%)	4.4	0.1	<0.001
Symptoms (%)	Vomits	75.3	11.4	<0.001
	Difficulty breathing	72.4	11.1	<0.001
	Abdominal pain	71.1	8.4	<0.001
	Sore throat	58.4	6.0	<0.001
	Nasal congestion	54.7	8.6	<0.001
	Diarrhea	54.7	7.6	<0.001
	Dry cough	41.9	6.6	<0.001
	Fever	41.0	10.4	<0.001
	Loss of taste	35.6	12.0	<0.001
	Loss of smell	34.0	11.9	<0.001
	Chill	33.4	8.4	<0.001
	Headache	23.6	3.1	<0.001
	Myalgia	25.2	5.4	<0.001
	Comorbidities	16.3	19.0	0.130
	Fatigue	12.5	2.6	<0.001
	Arterial hypertension	6.4	7.5	0.342
	Asymptomatic	30.4	84.5	<0.001

IQR = interquartile range.

**Table 2 viruses-15-00930-t002:** Commercial immunoassay concordance in discrepant samples from the study cohort (*n* = 164).

Manufacturer	Roche Concordance Kappa [95% CI]	Vircell IgG Concordance Kappa [95% CI]	Infection Concordance Kappa [95% CI]
Roche Elecsys^®^	-	−0.32 [−0.45–(−0.18)]	0.73 [0.58–0.87]
Vircell IgG	−0.32 [−0.45–(−0.18)]	-	−0.27 [−0.38–(−0.15)]
Diasorin IgG	0.58 [0.39–0.76]	−0.18 [−0.28–(−0.08)]	0.81 [0.68–0.94]
Euroimmun IgG	0.61 [0.39–0.82]	0.12 [0.04–0.20]	0.73 [0.56–0.89]

CI = confidence interval.

**Table 3 viruses-15-00930-t003:** Roche Elecsys^®^ and Vircell IgG performance among HCWs in the RT-PCR group from the study cohort (*n* = 425).

Immunoassay	RT-PCR Positive HCWs Group (*n* = 203)	RT-PCR Negative HCWs Group (*n* = 222)	RT-PCR Kappa [CI] (*n* = 425)	SARS-CoV-2 Infection Kappa [CI] (*n* = 425)
Roche Elecsys^®^-Positive	182 (89.7%)	67 (30.2%)	0.59 [0.51–0.66]	0.89 [0.85–0.94]
Roche Elecsys^®^-Negative	21 (10.3%)	155 (69.8%)		
Vircell IgG-Positive	180 (88.7%)	71 (32.0%)	0.56 [0.48–0.64]	0.84 [0.79–0.89]
Vircell IgG-Negative	23 (11.3%)	151 (68.0%)		
SARS-CoV-2 Infection	203 (100.0%)	67 (30.2%)	0.68 [0.62–0.75]	-

CI = confidence interval.

**Table 4 viruses-15-00930-t004:** Immunoassay performed to diagnose previous SARS-CoV-2 infection in the study cohort (*n* = 3550).

Manufacturer	Sensitivity	Specificity	Positive Predicted Value	Negative Predicted Value	Accuracy (95% CI)	Infection Concordance (Kappa, 95% CI)
Roche Elecsys^®^	94.8	99.8	99.1	99.0	99.3 (99.8–99.9)	0.96 (0.95–0.97)
Vircell IgG	92.9	95.3	77.8	98.6	96.9 (96.1–97.6)	0.82 (0.80–0.85)

CI = confidence interval.

**Table 5 viruses-15-00930-t005:** Roche Elecsys^®^ immunoassay performed to diagnose SARS-CoV-2 infection in the validation cohort (*n* = 1890).

Manufacturer	Sensitivity	Specificity	Positive Predicted Value	Negative Predicted Value	Accuracy (95% CI)	Infection Concordance (Kappa, 95% CI)
Roche Elecsys	95.3	99.7	98.6	99.1	96.9 (95.3–98.5)	0.96 (0.95–0.98)

CI = confidence interval.

## Data Availability

Data are available upon request from the corresponding author.

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
