# Peer review of "Validation of N Protein Antibodies to Diagnose Previous SARS-CoV-2 Infection in a Large Cohort of Healthcare Workers: Use of Roche Elecsys® Immunoassay in the S Protein Vaccination Era"

_viruses, 2023, doi:10.3390/v15040930_

Round 1
Reviewer 1 Report
To the Authors
In their article “Validation of N protein antibodies to diagnose SARS-CoV-2 infection in a large cohort of healthcare workers: Use of Roche Elecsys® immunoassay in the S protein vaccination era” the authors present a study of antibody titres in a large cohort of Healthcare Workers. Serum samples were taken in May 2020, before vaccinations became available. clinical characteristics were available for the whole study population. For ~1/7 of the ~3500 samples in the cohort PCR results were also available. The authors used two serological immunoassays from Roche and Vircell to assess antibody titres in all samples and in case they were discrepant, they used 2 more tests from Euroimmun and DiaSorin to determine a consensus on seropositivity. The authors compare assay performance between the Roche and Vircell assays based on an infection classifier and conclude that the Roche assay has better performance and can serve as a useful tool to classify infection in the population as large parts are vaccinated with Spike-specific vaccines and thus will not have Nucleocapsid specific antibodies unless infected. This is a relevant topic and while many studies with similar approaches have been conducted, it remains important research.
However, I have major concerns about the analysis and conclusion the authors present, which lead me to recommend the editor to reject the manuscript in its current form. These are listed below:
- In general, the authors make claims about the use of N-specific antibodies to determine breakthrough infections. While this is a reasonable thought, they only show data on samples before vaccination became available. The authors do not address this. They should also discuss current literature, which suggests that Nucleocapsid antibodies are less predictive of breakthroughs in vaccinated individuals compared to infected only. (https://doi.org/10.7326/M22-1300). In my opinion, the conclusion is not supported by the presented data and is too speculative to warrant the straightforward conclusion and choice of title, with the wording “…Use of Roche Elecsys immunoassay in the S protein vaccination era.”.
- The authors introduce a major bias in their analysis which invalidates large parts of their conclusion. The self-determined classifier for infection is used to evaluate the used assays, but in turn is based on their own results. For example, Table 3: The authors cannot correlate their infection classifier, which is defined in part by the two assays to classification of the serological immunoassays. They are biasing themselves heavily toward a good correlation. I suggest removing this column from table 3 (remove row SARS-CoV-2 infection and column SARS-CoV-2 infection kappa). The same is true for table 2 (column infection concordance kappa). The analysis in Table 4 falls victim to the same bias. PCR tests were only available for ~1/7 of the study population, so the remaining ~6/7 are solely classified as infected or uninfected based on the immunoassays, which of course results in very good values for specificity and sensitivity, since the two assays correlate well. The other assays are shown to correlate better with the Roche assay and therefore it is going to have a better performance compared to the Vircell assay in case of discordant results. The analysis of Table 4 cannot be performed with this data set and must be removed. Only comparing to PCR Test for infection is viable since it is an independent clinical metric for infection. Conclusions drawn from these analyses also have to be removed from the discussion. Using the available data to them, the authors can compare sensitivity and specificity of the serological assays within the group of PCR diagnosed individuals and can determine overall seropositivity for the whole cohort as well as perform the correlation between the different assays. They can use seropositivity or even their infection classifier to correlate with clinical characteristics as they do in the manuscript. However, they cannot discuss about assay performance with how they currently classify infected samples.
- PCRs were performed between February and May of 2020. The context of these PCRs is not completely disclosed in the text, and it is thus unclear whether a negative result means that there was no infection at all before sample collection or whether this was merely a snapshot. In the 425 individuals with PCR tests, many returned negative results meaning low viral load (between February and May), but showed antibodies (which were measured in May, potentially 3 months later). The PCR for these individuals was negative, but this does not exclude the possibility of a later infection and subsequent seroconversion. The lack of regular PCR testing in this cohort has likely hurt correlation of PCR with seroconversion, which I would expect to be quite good if the whole cohort was regularly tested and all infection events would have been caught. The authors should discuss these antibody titres in PCR negative samples. It contrasts with the presented pre-pandemic samples, where the Roche assay did not yield false positive results.
- The authors should give catalogue numbers for the used commercial assays
Author Response
General Statement:
“In their article “Validation of N protein antibodies to diagnose SARS-CoV-2 infection in a large cohort of healthcare workers: Use of Roche Elecsys® immunoassay in the S protein vaccination era” the authors present a study of antibody titres in a large cohort of Healthcare Workers. Serum samples were taken in May 2020, before vaccinations became available. clinical characteristics were available for the whole study population. For ~1/7 of the ~3500 samples in the cohort PCR results were also available. The authors used two serological immunoassays from Roche and Vircell to assess antibody titres in all samples and in case they were discrepant, they used 2 more tests from Euroimmun and DiaSorin to determine a consensus on seropositivity. The authors compare assay performance between the Roche and Vircell assays based on an infection classifier and conclude that the Roche assay has better performance and can serve as a useful tool to classify infection in the population as large parts are vaccinated with Spike-specific vaccines and thus will not have Nucleocapsid specific antibodies unless infected. This is a relevant topic and while many studies with similar approaches have been conducted, it remains important research.”
Reply: We appreciate the encouraging and valuable comments made by the reviewer. A point-by-point response to his concerns can be found below.
Major comments
- Point 1: “In general, the authors make claims about the use of N-specific antibodies to determine breakthrough infections. While this is a reasonable thought, they only show data on samples before vaccination became available. The authors do not address this. They should also discuss current literature, which suggests that Nucleocapsid antibodies are less predictive of breakthroughs in vaccinated individuals compared to infected only. (https://doi.org/10.7326/M22-1300). In my opinion, the conclusion is not supported by the presented data and is too speculative to warrant the straightforward conclusion and choice of title, with the wording “…Use of Roche Elecsys immunoassay in the S protein vaccination era.”.
Reply: We thank the reviewer for drawing our attention to this issue. In the study we analyzed the antibody response against the N protein prior to vaccination to avoid confounding factors regarding the ability of these antibodies to detect infection. But we do agree on the fact that in our work the antibody response against N protein after the appearance of vaccines against the S protein is not analyzed.
During the pandemic we made 3 cross sections studies to measure the antibody response against SARS-CoV-2: May 2020 (the present study), December 2020 (also pre-vaccination) and June 2021, in which the vast majority of HCWs had been vaccinated. For the 2021 cross section we have performed serologies against S protein (ref 24) and N protein, on this occasion only by the Roche Elecsys® immunoassay. To carry out the required analysis, we have added to the manuscript the data of infected HCWs who were not in the study cohort, comparing the immunoassay to detect antibodies against the N protein with the RT-PCR. (Lines 244-253, and Table 5)
- Point 2: “The authors introduce a major bias in their analysis which invalidates large parts of their conclusion. The self-determined classifier for infection is used to evaluate the used assays, but in turn is based on their own results. For example, Table 3: The authors cannot correlate their infection classifier, which is defined in part by the two assays to classification of the serological immunoassays. They are biasing themselves heavily toward a good correlation. I suggest removing this column from table 3 (remove row SARS-CoV-2 infection and column SARS-CoV-2 infection kappa). The same is true for table 2 (column infection concordance kappa). The analysis in Table 4 falls victim to the same bias. PCR tests were only available for ~1/7 of the study population, so the remaining ~6/7 are solely classified as infected or uninfected based on the immunoassays, which of course results in very good values for specificity and sensitivity, since the two assays correlate well. The other assays are shown to correlate better with the Roche assay and therefore it is going to have a better performance compared to the Vircell assay in case of discordant results. The analysis of Table 4 cannot be performed with this data set and must be removed. Only comparing to PCR Test for infection is viable since it is an independent clinical metric for infection. Conclusions drawn from these analyses also have to be removed from the discussion. Using the available data to them, the authors can compare sensitivity and specificity of the serological assays within the group of PCR diagnosed individuals and can determine overall seropositivity for the whole cohort as well as perform the correlation between the different assays. They can use seropositivity or even their infection classifier to correlate with clinical characteristics as they do in the manuscript. However, they cannot discuss about assay performance with how they currently classify infected samples.”
Reply: The SARS-CoV-2 infection criteria proposed in our study includes the positivity of 2 serological tests and/or the positivity of the RT-PCR. Given that, as the reviewer rightly points out, 6/7 of the study population did not have a RT-PCR performed, thus the infection criterion is mostly met by the presence of 2 positive serological tests. The question to ask would be is the detection of antibodies against 2 different SARS-CoV-2 antigens (N protein on the Roche assay and peptides from S and N proteins on Vircell assay) indicative of the presence of infection? The presence of antibodies against at least 2 different SARS-CoV-2 antigens is linked to a previous infection by this virus, therefore, we believe that the infection criteria is correct. Does it make sense to establish the correlation of a serological test that is part of the criteria to establish the infection? In our opinion, it does. Among the 563 infected HCWs, 203 had a positive RT-PCR and, therefore, are considered infected and consequently, 36.1% of HCWs have been classified regardless of the serology result. The remaining 63.9% have been classified based on two positive tests to give greater certainty to the diagnosis of previous infection. It is true that there is a good correlation between Roche and Vircell but despite this good correlation, the analysis with the presence of infection allows us to discriminate which serological test is better to diagnose previous infection and, therefore, this observation already makes the analysis worthwhile. Even so, we appreciate the reviewer's suggestion to remove the SARS-CoV-2 infection row since it does not add value to Table 3. For the reasons described above, we will keep the correlation with infection column in Tables 2 and 4.
We agree with the reviewer that RT-PCR is the gold standard technique for the diagnosis of infection and, in fact, its positivity regardless of the serology result, is an infection criterion itself. However, we do not agree with two positive serological tests not being an indicative of previous infection as the presence of antibodies is due to a cause. The specificity for both serological tests has been studied with samples from blood donors prior to the pandemic and a cut-off point that gave us a specificity of 98% has been stablished Taking this data into account, we do not find other explanation to the positivity in two serological tests than having a previous SARS-CoV-2 infection.
- Point 3: “PCRs were performed between February and May of 2020. The context of these PCRs is not completely disclosed in the text, and it is thus unclear whether a negative result means that there was no infection at all before sample collection or whether this was merely a snapshot. In the 425 individuals with PCR tests, many returned negative results meaning low viral load (between February and May), but showed antibodies (which were measured in May, potentially 3 months later). The PCR for these individuals was negative, but this does not exclude the possibility of a later infection and subsequent seroconversion. The lack of regular PCR testing in this cohort has likely hurt correlation of PCR with seroconversion, which I would expect to be quite good if the whole cohort was regularly tested and all infection events would have been caught. The authors should discuss these antibody titres in PCR negative samples. It contrasts with the presented pre-pandemic samples, where the Roche assay did not yield false positive results.”
Reply: We share the referee's concern about the lack of regular PCR testing and how this has most likely impacted the correlation between PCR and seroconversion. We will expand the discussion realted with this topic. (Lines 291-295)
- Point 4: “The authors should give catalogue numbers for the used commercial assays.”
Reply: We have added in Materials and Methods the catalogue numbers of the commercial immunoassays used in the study. (Lines 129-144)

Reviewer 2 Report
I read and reviewed with interest the manuscript entitled “Validation of N protein antibodies to diagnose SARS-CoV-2 infection in a large cohort of healthcare workers: Use of Roche Elecsys® immunoassay in the S protein vaccination era” by Juan Francisco Delgado et al. In my opinion, in its present form the paper is not suitable for publication in Viruses. While the findings are very relevant, the cohort of studied patients is very large, the timing of blood collection is relevant to avoid S-reactivity confusion and the approach adequate; the manuscript does not present results with clarity and English must be edited for correctness extensively. Redesign of the manuscript would certainly enhance the relevance of their findings. Specifically:
Major comments:
- The Introduction is verbose, it needs to be edited. There is information in the discussion that is very relevant as introduction. For instance the fact that just a fraction of symptomatic patients were studied by RT-PCR Is very important. As a matter of fact, the paper would benefit from a figure where the distribution of patients is depicted. This would help understand how the data presented in the different tables is important and relates globally to the findings.
- The reactivity of the immunoassays need to be spelt out in the introduction and a description of their criteria for positivity displayed in tables. Are the immunoassays quantitative? If so, a bar graphic (per immunoassay) is in order. Is the RT-PCR quantitative, if so a description of their positivity is also in order.
- In table 1, there is a line it read” Days after (the) onset (of) symptoms”, is this the time of blood-collection? If so it is very relevant and should be described as such. In the same table “Loss of taste” is called ageusia and “loss of smell” is called anosmia. In that table please delete the line “smoker” the study is not powered to find differences in COVID susceptibility and it reads like smoking protects against COVID-19 in HCW. Isn’t hypertension a comorbidity?
- Please define “Infection concordance” How is it calculated?
- English need to be edited for correctness. An example, in lines 62 and 63 it reads: “Both S and N proteins are vital structural proteins with good immunogenicity antibodies that are elicited in most patients [15].” It is not correct. It should read: “Both viral S and N proteins are major structural proteins and highly immunogenic. Therefore, most patients develop antibodies against them”. Another example, the world population has been vaccinated against the spike (S) protein; the world population has been vaccinated against COVID-19 using the spike (S) protein as antigen.
Minor comments:
1. Too many to enumerate. For instance, every HCW is an adult per definition (line 98).
Author Response
General Statement:
“I read and reviewed with interest the manuscript entitled “Validation of N protein antibodies to diagnose SARS-CoV-2 infection in a large cohort of healthcare workers: Use of Roche Elecsys® immunoassay in the S protein vaccination era” by Juan Francisco Delgado et al. In my opinion, in its present form the paper is not suitable for publication in Viruses. While the findings are very relevant, the cohort of studied patients is very large, the timing of blood collection is relevant to avoid S-reactivity confusion and the approach adequate; the manuscript does not present results with clarity and English must be edited for correctness extensively. Redesign of the manuscript would certainly enhance the relevance of their findings.”
Reply: We thank the reviewer for his valuable comments, which we will address in the following sections.
Major comments:
- Point 1: “The Introduction is verbose, it needs to be edited. There is information in the discussion that is very relevant as introduction. For instance the fact that just a fraction of symptomatic patients were studied by RT-PCR Is very important. As a matter of fact, the paper would benefit from a figure where the distribution of patients is depicted. This would help understand how the data presented in the different tables is important and relates globally to the findings.”.
Reply: We thank the reviewer for drawing our attention to the general understanding of the article which can be improved through a figure that clarifies the tests that have been carried out in HCWs. (Line 115)
- Point 2: “The reactivity of the immunoassays need to be spelt out in the introduction and a description of their criteria for positivity displayed in tables. Are the immunoassays quantitative? If so, a bar graphic (per immunoassay) is in order. Is the RT-PCR quantitative, if so a description of their positivity is also in order.”.
Reply: We thank the reviewer for pointing out this issue. We have added the reactivity of the immunoassays in the introduction and in Table S2. (Line 87-90). The immunoassays are semi-quantitative, therefore, we have performed boxplot figures grouping the results by the presence of infection and the serological test used. Furthermore, we have added the cut-off values for the immunoassays (Lines 135-136, 138). The RT-PCR is not quantitative; it’s just qualitative informing positive or negative, also without informing the threshold cycle (Ct).
- Point 3: “In table 1, there is a line it read” Days after (the) onset (of) symptoms”, is this the time of blood-collection? If so it is very relevant and should be described as such. In the same table “Loss of taste” is called ageusia and “loss of smell” is called anosmia. In that table please delete the line “smoker” the study is not powered to find differences in COVID susceptibility and it reads like smoking protects against COVID-19 in HCW. Isn’t hypertension a comorbidity?”.
Reply: The line in Table 1 that refers to the days after the onset of symptoms refers to the time between the onset of symptoms and the blood-collection. We have modified Table 1 according to this point, and we have also corrected the name of the symptoms mentioned. Regarding the Smoker line, what the table shows is that the frequency of smokers in the group of infected HCWs is lower than in the group of non-infected HCWs, and this difference in frequency is significant. This is what we observed and in the discussion we address this topic, in which although being a smoker induces a more serious disease, the presence of smokers in infected patients is lower than in non-infected ones, being a paradoxical fact. We do not understand why this fact has to be removed from Table 1. The last question is striking because arterial hypertension is a risk factor for SARS-CoV-2 infection, but we have not found this association in our cohort.
Point 4: “Please define “Infection concordance” How is it calculated?”.
Reply: We use the term concordance with infection to refer to the level of agreement between serology positivity and infection. To analyze this parameter, we use Cohen's kappa coefficient.
Point 5: “English need to be edited for correctness. An example, in lines 62 and 63 it reads: “Both S and N proteins are vital structural proteins with good immunogenicity antibodies that are elicited in most patients [15].” It is not correct. It should read: “Both viral S and N proteins are major structural proteins and highly immunogenic. Therefore, most patients develop antibodies against them”. Another example, the world population has been vaccinated against the spike (S) protein; the world population has been vaccinated against COVID-19 using the spike (S) protein as antigen.”.
Reply: Since we are not native English speakers, we have sent a draft of the manuscript to American Manuscript Editors for proofreading. We have sent the certificate of edition to the journal. Nevertheless, we will change theses sentences to make them more comprehensive. (Lines 61-62). Additionaly, we have send the manuscript to another English reviewer.
Minor comments:
- Point 1: “Too many to enumerate. For instance, every HCW is an adult per definition (line 98).”
Reply: According with the previous point we have made the modification. (Line 99)

Reviewer 3 Report
Juan F Delgado et al. evaluated the Roche Elecsys anti SARS-CoV-2 N-protein immunoassay using plasma and nasopharyngeal samples for healtcare workers (HCW).
Anti-N antibodies detection is useful to diagnosis a previous SARS-CoV-2 infection. The gold standard to diagnosis current SARS-CoV-2 infection is the NP RT-PCR.
According to me, this study would be interesting to validate the Roche Elecsys immunoassay for the detection of previous SARS-CoV-2 infection.
Some important points should be clarify:
- Time of samples collection (NP samples and blood samples) and clinical symptoms are not clear.
Were the NP and the serum samples collected at the same time?
It is now well documented that serological tests should be used 3 weeks after SARS-CoV-2 exposure, to have acceptable sensitivity. However, NP samples for RNA detection have to be collected 3 to 7 days post-exposure.
The authors detailed clinical symptoms of the HCW, were these datas collected during the samples and referred to symptoms presents at collection time or before? Symptoms should appear 3 to 7 days after contamination. If the symptoms are described during the right blood collection (3 weeks after exposure), we cannot be sure that the symptoms are specific to SARS-CoV-2 infection.
- Were the HCW exposed to patients known to be SARS-CoV-2 positive? Or did the HCW have NP samples collected because they suspected SARS-CoV-2 infection due to their symptoms?
- 3,550 HCW were enrolled in this study from 6 to 29 may 2020; however lines 164-165 it is mentioned that 425 NP samples were tested between February and May 2020. These sentences are not clear. Did the remaining 3125 HCW have NP samples collected or not?
All these points should be clarify in a new version of the manuscript.
Others points need to be clarified:
- Introduction:
o Lines 40-43: this is very general and known.
o Line 79: please indicate what type of vaccines you refer to.
- Table 1:
o Female/male ratio: 3.8 ? is that a percentage ?
o Hospitalization (%): for COVID-19 ?
- Materials:
o Line 106: could you date SARS-CoV-2 infection? It is possible using RT-PCR because the infection should be around the date of the positive test, however for serological immunoassays it is difficult to date the infection.
o When you refer to manufacter’s instructions please add a reference.
o Commercial immunoassays: please specify if the test detects N or S protein or both? and what result you used for the analysis.
o Line 128: were the healthy donors HCW too ? Please had informations about their characteristics (age, sexe,…). Did you use multivariate analysis to compare these two groups (statistical analysis) ? If not, this need to be discuss in the discussion
- Results
o Line 147: please add a reference for manufacturer cutoff.
o 3.2 immunoassays performance: you could detailed the number of samples that were positive with Roche and Diasorin + Euroimmun assays compared to those positive with Vircell and Diasorin + Euroimmun, in order to compare Roche and Vircell immunoassays.
o Line 176: 563 HCW among 3,550 HCW ? please clarify
- Discussion
o Roche immunoassay detect total anti-N antibodies whereas Vircell only detect type G Ig. If you use this explanation, Roche Elecsys would detect more antibodies that Vircel, or it is the contrary.
o Line 244: what cutoff was chosen for the analysis ? In this sentence you said that you did not use Vircell recommanded cutoff, however line 147 it is written that “manufacturer’s recommended index cutoff (6)” was used.
o Lines 250-267: did you use multivariate analysis ? if not, the two populations can’t be compared like that. This should be discussed.
o Lines 268-278: it is difficult to compared clinical parameters in the positive serological test group and in the RT PCR positive groups as NP samples are collected during the symptoms period whereas blood samples for serological diagnosis should be collected 3 weeks later.
- Conclusion
o Please mention “previous” SARS-CoV-2 infection when you refer to serological test.
- Correlations could be illustrated by dot plot graphs.
Author Response
General Statement:
“Juan F Delgado et al. evaluated the Roche Elecsys anti SARS-CoV-2 N-protein immunoassay using plasma and nasopharyngeal samples for healtcare workers (HCW). Anti-N antibodies detection is useful to diagnosis a previous SARS-CoV-2 infection. The gold standard to diagnosis current SARS-CoV-2 infection is the NP RT-PCR. According to me, this study would be interesting to validate the Roche Elecsys immunoassay for the detection of previous SARS-CoV-2 infection.”
Reply: We thank the referee for their revision of our work. Below we explain how we have addressed their concerns in our revised version of the manuscript.
Important points:
- Point 1: “Time of samples collection (NP samples and blood samples) and clinical symptoms are not clear. Were the NP and the serum samples collected at the same time? It is now well documented that serological tests should be used 3 weeks after SARS-CoV-2 exposure, to have acceptable sensitivity. However, NP samples for RNA detection have to be collected 3 to 7 days post-exposure. The authors detailed clinical symptoms of the HCW, were these datas collected during the samples and referred to symptoms presents at collection time or before? Symptoms should appear 3 to 7 days after contamination. If the symptoms are described during the right blood collection (3 weeks after exposure), we cannot be sure that the symptoms are specific to SARS-CoV-2 infection.”
Reply: We agree with the referee that this part of the sample collection may lead to confusion. In May 2020, blood samples were collected. However, the collection of samples to perform the RT-PCR was carried out in the period from March to May 2020, depending on the symptoms of the HCWs and the availability of PCR reagents. This is a limitation of the study which translates into a bad correlation between RT-PCR and infection, due to the fact that the samples were obtained outside the acute period of infection leading to a high number of false negative results, topic address in the discussion.
The symptoms were recorded at the same time of blood sample collection, but they refer to the entire period between March and May. This is a limitation of the study and it can induce a bias due to the fact that the symptoms registered were the symptoms that HCWs could remember, although the maximum period of time would be 3 months. The possibility of forgetting a symptom, despite the fact that it is plausible, is low. Another limitation would be that HCWs who have experienced an asymptomatic SARS-CoV-2 infection may have reflected symptoms due to another pathological process. We have added a better explanation of how we have obtained the clinical data (lines 101-102)
- Point 2: “Were the HCW exposed to patients known to be SARS-CoV-2 positive? Or did the HCW have NP samples collected because they suspected SARS-CoV-2 infection due to their symptoms?”
Reply: The indication for performing the RT-PCR test were the 2 described by the reviewer in the question, being a close contact of a patient with SARS-CoV-2 infection and the presence of symptoms compatible with infection.
- Point 3: “3,550 HCW were enrolled in this study from 6 to 29 may 2020; however lines 164-165 it is mentioned that 425 NP samples were tested between February and May 2020. These sentences are not clear. Did the remaining 3125 HCW have NP samples collected or not?”
Reply: The RT-PCR has only been performed on 425 HCWs, therefore, 3125 HCWs did not have a RT-PCR performed and we do not have the NP samples.
Other points:
- Point 1: “Introduction: Lines 40-43: this is very general and known. Line 79: please indicate what type of vaccines you refer to.”
Reply: In accordance with the reviewer's suggestion, we have added the mRNA vaccines against which an antibody response has been obtained. (Line 78)
- Point 2: “Table 1: Female/male ratio: 3.8 ? is that a percentage? Hospitalization (%): for COVID-19 ?.”
Reply: The female/male ratio of 3.8 refers to the fact that for every man there are 3.8 women in the study, therefore, it is not a percentage.
The percentage of hospitalized HCWs does not only refer to that induced by SARS-CoV-2 infection, for this reason there were 3 HCWs in the non-infected group who have required hospitalization due to the presence of other pathological processes.
- Point 3: “Materials: Line 106: could you date SARS-CoV-2 infection? It is possible using RT-PCR because the infection should be around the date of the positive test, however for serological immunoassays it is difficult to date the infection. When you refer to manufacter’s instructions please add a reference. Commercial immunoassays: please specify if the test detects N or S protein or both? and what result you used for the analysis. Line 128: were the healthy donors HCW too ? Please had informations about their characteristics (age, sexe,…). Did you use multivariate analysis to compare these two groups (statistical analysis) ? If not, this need to be discuss in the discussion.”
Reply: According to the reviewer's comment, we could only date the infection in the 203 HCWs in which we have a positive RT-PCR for SARS-CoV-2, while for the rest of the infected HCWs it would be more difficult, with the symptom onset date.
Supplementary Table 2 explains the antigens recognized by each assay and the methodology for the detection of antibodies. In addition, we have added the cut-off point used for the 2 immunoassays in materials and methods. (Lines 136, 138)
The healthy donors included in the study are not HCWs, they come from the Banc de Sang i Teixits. We have added in material and methods section the source of these sera and their demographic characteristics. (Lines 147-150).
We do not understand what the reviewer is referring to if we have used a multivariate analysis to compare these 2 groups. The analyzes performed to determine the performance of the serological immunoassays were univariate. Analysis of the performance of immunoassays is based on the comparison between the qualitative result provided by an immunoassay with respect to the presence or absence of infection. This comparison is not influenced by the qualitative result of the other immunoassay, therefore, we understand that the univariate analysis is correct, since there are 2 variables that are measuring the same parameter (the presence of antibodies against SARS-CoV-2) and have no relationship or influence on each other. We have added in statistical analysis that we used univariate analysis. (Line 158)
- Point 4: “Results Line 147: please add a reference for manufacturer cutoff. 3.2 immunoassays performance: you could detailed the number of samples that were positive with Roche and Diasorin + Euroimmun assays compared to those positive with Vircell and Diasorin + Euroimmun, in order to compare Roche and Vircell immunoassays. Line 176: 563 HCW among 3,550 HCW ? please clarify.”
Reply: We have added the cut-off point for the immunoassays in lines 136 and 138.
We have added the data from the discrepant samples in the immunoassays suggested by the reviewer. (Lines 176-179)
We have added the clarification proposed by the reviewer: 563 HCW among 3,550 HCWs. (Line 213)
- Point 5: “Discussion Roche immunoassay detect total anti-N antibodies whereas Vircell only detect type G Ig. If you use this explanation, Roche Elecsys would detect more antibodies that Vircel, or it is the contrary. Line 244: what cutoff was chosen for the analysis ? In this sentence you said that you did not use Vircell recommanded cutoff, however line 147 it is written that “manufacturer’s recommended index cutoff (6)” was used. Lines 250-267: did you use multivariate analysis ? if not, the two populations can’t be compared like that. This should be discussed. Lines 268-278: it is difficult to compared clinical parameters in the positive serological test group and in the RT PCR positive groups as NP samples are collected during the symptoms period whereas blood samples for serological diagnosis should be collected 3 weeks later.”
Reply: For the Roche immunoassay, as the reviewer rightly comments, the cutoff recommended by the manufacturer is the one used in the study. However, for the Vircell immunoassay we have used our own cutoff after analyzing the specificity of the immunoassay with samples from healthy donors, and getting poor specificity results with the manufacturer’s recommended cut-offs. Therefore, in the analysis we used Roche manufacture’s cutoff and our own cutoff for Vircell.
We have established an infection criterion for the study population and based on this criterion we have analyzed the clinical parameters collected in the survey carried out by the HCWs in the period March-May 2020. We understand that with the clarification exposed previously explaining better the method of collecting clinical data, there were no differences between the group that has been classified as infected using RT-PCR criterion and those that have been classified as infected exclusively by the 2 positive serological tests criterion.
The analysis of the symptoms has been discussed previously and in our opinion, with some limitations due to the chosen methodology, it would be possible to compare the symptoms in the group of patients diagnosed by RT-PCR and those diagnosed exclusively by serological tests.
- Point 6: “Conclusion Please mention “previous” SARS-CoV-2 infection when you refer to serological test.”
Reply: We totally agree with the referee. We have changed the sentence (Lines 355-356)
- Point 7: “Correlations could be illustrated by dot plot graphs.”
Reply: We appreciate the reviewer's comment and have added a supplementary figure S1 in which a dot plot is made to show the correlation between the Roche Elecsys and Vircell results.

Round 2
Reviewer 1 Report
The authors have included a validation cohort with substantial sample numbers. The infection classifier for this cohort is not based on any serological results, but solely on PCR results. This cohort is well suited to describe Sensitivity and Specificity of the Roche assay, which the authors do in the new table 5. They have furthermore thereby provided samples post vaccination, which warrants the choice of their title.
However, the authors insist on keeping the infection classifier of the original cohort for describing assay sensitivity and specificity. I cannot stress enough that this is a technically incorrect analysis. I fully agree with the infection classifier itself as either PCR indicating current infection or antibodies indicating past infections. This classifier is well suited to perform analysis of the clinical parameters. However, since the classifier for a major part of the study cohort depends on the Roche assay, it may not be used to evaluate that same assay, no matter how sound the classifier is scientifically.
To help explain my point consider the following: You have 1000 participants and for 100 you have PCR results indicating that 50 out of the 100 have been infected. Instead of performing a seroassay you toss a coin for each participant. For the 900 participants without PCR you use the coin toss to indicate infection, resulting in 450 infected and 450 uninfected participants (the most likely outcome). In the 50 participants with negative PCR you call 25 positive and in the 50 participants with positive results you call 25 negative, by chance. Overall, your coin toss called 500 participants positive and overlaps with 475 of the hybrid infection classifier. The resulting sensitivity for a coin toss vs infection is 95%. Does this make a coin toss a good classifier for SARS-CoV-2 infection? Of course not.
The fact that the authors used multiple assays and that seropositivity correlates well with previous infection means that the numbers are probably close to the truth, which is indicated by the similarity to table 5, but this does not change the fact the authors analysis is incorrect on a technical level and the numbers for sensitivity and specificity have no merit.
The authors should remove Table 4 entirely and use the new table 5 instead to indicate assay performance. The authors should also remove the Infection classifier from Tables 2 and 3 as the numbers again have no merit. Table 1 is perfectly fine to keep the infection classifier, as there is no interaction between the classifier and assay performance here. Only then can this manuscript be accepted for publication in my opinion.
Author Response
We appreciate the encouraging and valuable comments made by the reviewer. We thank the reviewer for his comments on the changes that we have made in the manuscript.
As reviewer comments, we both fully agree with the infection classifier itself as PCR indicating current infection or antibodies indicating past infections. For both, the criterion is adequate for the study of clinical characteristics, but we disagree that it can be used for the analysis of the performance of immunoassays. We understand the reviewer's concern, because although at a scientific level it makes perfect sense, at a mathematical level making comparisons when the variable is part of the classification criteria could generate a bias. Considering this issue, we have added in materials and methods that the performance analysis of the serology could induce a bias, but we have tried to reduce it by classifying as infection if a HCW has obtained two positive serology results and in case of discrepancy adding two additional serological tests. In other words, we are not classifying a HCW as infected based on the PCR result and the serological result obtained by Roche, or the PCR result and the serological result obtained by Vircell, otherwise the bias introduced would be large. Rather, we have classified based on the result of the PCR and the joint assessment of the Roche and Vircell serology result. (Lines 123-131; 311-313)
We sincerely believe that in this way the bias that can be induced when studying the performance of a variable that is part of the classification criterion is largely mitigated. And also, as you well recommended in the previous review, we have added a validation cohort that has already been vaccinated and whose infection classification criterion was the PCR result because the availability of reagents and the indication of the test was done correctly.
The introduction of these correction factors to reduce the bias has shown to be useful since the results obtained in the validation cohort were similar to those of the study cohort, which could potentially be biased. Therefore, given the inclusion of this validation cohort and the replication of the results with respect to the study cohort, we consider that it is possible to show the results obtained by reporting the possible bias that we may be introducing when analyzing the performance of the serological tests.

Reviewer 2 Report
I have found the revised manuscript much improved. it reads better and the results support authors claims. As such I recommend publication in its present form.
Author Response
We thank the reviewer for his valuable that has allowed us to improve this manuscript.

Reviewer 3 Report
The authors modified the manuscript, but I think it has still a lack of originality. I agree that the cohort is interesting, however in my opinion the main message and the titles are not correct. N protein antibodies immunoassay can be used to diagnose previous SARS-CoV-2 infection only.
If the aim of the study is really to validate anti-N Roche Elecsys immunoassay to diagnosis SARS-CoV-2 infection, the authors should evaluate that by comparing the results to RT-PCR results. However, Table 4, presents Sensitivity, Specificity, PPV, NPS, Accuracy for the 3,550 whereas all these individuals did not have RT-PCR results. In my opinion, only the table 5 show real performance analysis of the Roche Elecsys immunoassay. If this is the aim of the study, it should be modified
Author Response
We thank the referee for his revision of our work and his comments. The reviewer expresses his concern about the possibility of analyzing the performance of serological tests in the study cohort when not all HCWs in it have an RT-PCR result. It is widely known that having a positive serology against a pathogen is indicative of having suffered a previous infection by that pathogen. Therefore, using the result of a serological test as a criterion to classify a subject as infected is theoretically correct. It is true that the gold standard diagnostic technique for SARS-CoV-2 infection is RT-PCR, which in the case of the study cohort could only be performed on 425 HCWs. The vast majority of HCWs reporting symptoms the 25th percentile of onset of symptoms was more than 41 days, therefore serological tests in that time window could have a great performance to detect previous infection. So, from a theoretical point of view, we could study the performance of the serological tests to detect previous SARS-CoV-2 infection. In this sense, we have added in Table 4 “previous infection” since it was the analysis that we have carried out, as well as in the manuscript title and conclusion. (Lines 3, 35, 222, and 354)
